# Genome-Wide Association Studies of Estimated Fatty Acid Desaturase Activity in Serum and Adipose Tissue in Elderly Individuals: Associations with Insulin Sensitivity

**DOI:** 10.3390/nu10111791

**Published:** 2018-11-17

**Authors:** Matti Marklund, Andrew P. Morris, Anubha Mahajan, Erik Ingelsson, Cecilia M. Lindgren, Lars Lind, Ulf Risérus

**Affiliations:** 1The George Institute for Global Health, University of New South Wales, Sydney, NSW 2042, Australia; 2Department of Public Health and Caring Sciences, Clinical Nutrition and Metabolism, Uppsala University, 751 22 Uppsala, Sweden; ulf.riserus@pubcare.uu.se; 3Department of Biostatistics, University of Liverpool, Liverpool L69 3GL, UK; a.p.morris@liverpool.ac.uk; 4The Wellcome Trust Centre for Human Genetics, Oxford OX3 7BN, UK; anubha@well.ox.ac.uk (A.M.); celi@broadinstitute.org (C.M.L.); 5Department of Medicine, Division of Cardiovascular Medicine, Stanford University School of Medicine, Stanford, CA 94305, USA; eriking@stanford.edu; 6Stanford Cardiovascular Institute, Stanford University, Stanford, CA 94305, USA; 7Stanford Diabetes Research Center, Stanford University, Stanford, CA 94305, USA; 8Department of Medical Sciences, Molecular Epidemiology, Uppsala University, 751 85 Uppsala, Sweden; 9Li Ka Shing Centre for Health Information and Discovery, The Big Data Institute, University of Oxford, Oxford OX3 7LF, UK; 10Department of Medical Sciences, Cardiovascular Epidemiology, Uppsala University, 751 85 Uppsala, Sweden; lars.lind@medsci.uu.se

**Keywords:** fatty acid, desaturase, Genome-wide association study (GWAS), Insulin sensitivity, adipose tissue, cholesterol ester

## Abstract

Fatty acid desaturases (FADS) catalyze the formation of unsaturated fatty acids and have been related to insulin sensitivity (IS). FADS activities differ between tissues and are influenced by genetic factors that may impact the link to IS. Genome-wide association studies of δ-5-desaturase (D5D), δ-6-desaturase (D6D) and stearoyl-CoA desaturase-1 (SCD) activities (estimated by product-to-precursor ratios of fatty acids analyzed by gas chromatography) in serum cholesterol esters (*n* = 1453) and adipose tissue (*n* = 783, all men) were performed in two Swedish population-based cohorts. Genome-wide significant associated loci were evaluated for associations with IS measured with a hyperinsulinemic euglycemic clamp (*n* = 554). Variants at the *FADS1* were strongly associated with D5D in both cholesterol esters (*p* = 1.9 × 10^−70^) and adipose tissue (*p* = 1.1 × 10^−27^). Variants in three further loci were associated with D6D in cholesterol esters (*FADS2*, *p* = 3.0 × 10^−67^; *PDXDCI*, *p* = 4.8 × 10^−8^; and near *MC4R*, *p* = 3.7 × 10^−8^) but no associations with D6D in adipose tissue attained genome-wide significance. One locus was associated with SCD in adipose tissue (*PKDL1*, *p* = 2.2 × 10^−19^). Genetic variants near *MC4R* were associated with IS (*p* = 3.8 × 10^−3^). The *FADS* cluster was the main genetic determinant of estimated FADS activity. However, fatty acid (FA) ratios in adipose tissue and cholesterol esters represent FADS activities in separate tissues and are thus influenced by different genetic factors with potential varying effects on IS.

## 1. Introduction

Fatty acid desaturases (FADS) catalyze the formation of mono- and polyunsaturated fatty acid and thus influence the fatty acid (FA) composition of the blood stream and in adipose tissue (AT). The major human FADS include δ-5-desaturase (D5D), δ-6-desaturase (D6D) and δ-9-desaturase or stearoyl-CoA desaturase-1 (SCD). While SCD synthesizes monounsaturated FA from saturated FA, D5D and D6D catalyze the formation of polyunsaturated FA. Since in vivo measurement of enzyme activity can be challenging, ratios of substrate and product concentrations are commonly used to estimate activities of SCD, D6D and D5D [1].

Both circulating fatty acid composition and estimated FADS activities have previously been associated with insulin sensitivity (IS) and incidence of type 2 diabetes (T2D), although the mechanisms underlying these relationships have not been fully determined [2]. It has been suggested that the effects of FADS on IS are mediated by alterations in FA compositions. This could lead to effects on cell membranes (influencing insulin receptor binding and affinity, translocation of glucose transporters and intercellular signaling) and altered levels of polyunsaturated fatty acids (PUFA) which function as ligands for a variety of transcription factors [3]. In addition, polymorphisms in the genes encoding for FADS have been was associated with fasting glucose levels and estimated β-cell function in a large-scale meta-analysis of genome-wide association studies (GWAS) [4]. 

Genome-wide associations of FADS activity estimated in the circulation have been reported previously and demonstrated that not only variants in desaturase encoding genes are associated with estimated FADS activity [5,6]. However, no prior GWAS have reported associations between genotype and activities of SCD, D5D, or D6D estimated in AT. Furthermore, previous studies investigating relationships between *FADS* polymorphisms and IS have relied on indirect measurements (e.g., fasting glucose and insulin) instead of gold standard methodology, such as the hyperinsulinemic euglycemic clamp. 

The aim of the present study was to perform GWAS of FA metabolizing enzymes in AT and serum cholesterol esters (CE) in participants of the Uppsala Longitudinal Study of Adult Men (ULSAM) and the Prospective Investigation of the Vasculature in Uppsala Seniors (PIVUS). Subsequently, analyses were performed to investigate relationships between desaturase-associated genetic variants and cardiometabolic risk factors, including IS assessed by hyperinsulinemic euglycemic clamp.

## 2. Materials and Methods 

### 2.1. Study Samples

Details about The Uppsala Longitudinal Study of Adult Men (ULSAM) are available in previous publications [7] and online at http://www.pubcare.uu.se/ulsam/. In brief, at the first collection time-point, all 50-year-old men living in Uppsala County, Sweden, 1970–74, were invited. The present study includes individuals at the re-examination undertaken from August 1991 to May 1995 at the approximate age of 71 years, where 1221 out of 1681 invited individuals participated (73% of those still alive and living in Uppsala). For the present study, we excluded individuals with missing microarray genotyping data (*n* = 5), failing sample quality control (QC) (*n* = 37), or missing all estimates of FADS activity (*n* = 615, cholesterol esters; *n* = 396, adipose tissue).

A detailed study description of The Prospective Investigation of the Vasculature in Uppsala Seniors (PIVUS) has been published previously [8] and additional information can be found at http://www.medsci.uu.se/pivus/pivus.htm. In brief, all 70-year-old individuals living in Uppsala County, Sweden, between April 2001 and June 2004 were eligible for the study, out of which 2025 randomly selected subjects were invited. In total, 1016 subjects (50% women) participated and were examined within one month of their 70th birthday to standardize for age. For the present study, we excluded individuals with missing microarray genotyping data (*n* = 34), failing sample QC (*n* = 33), or missing all outcome measurements (*n* = 60). 

Both the ULSAM and PIVUS studies were approved by the ethics committee of Uppsala University and all participants provided written informed consent.

### 2.2. Assessments of Fatty Acid Composition and Enzyme Activities

Procedures for measurements of fatty acid composition in ULSAM and PIVUS have previously been described in detail [9,10]. Briefly, FA were measured in CE from blood samples drawn after an overnight fast, during which both medication and smoking were disallowed. A hexane-isopropanol solution was used to extract serum, from which cholesterol esters were separated by thin-layer chromatography followed by inter-esterification with acidic methanol. Free cholesterol was removed by aluminum oxide to avoid contamination of the column. The relative proportion of methylated FA was determined by gas chromatography (25-m NB-351 silica capillary column) with a coefficient of variation <5.0%. In addition, FA in subcutaneous AT were analyzed in ULSAM as previously described [10]. An estimate of enzyme activity was calculated as the product-to-substrate ratio; 20:4n−6/20:3n−6 for D5D, 18:3n−6/18:2n−6 for D6D and 16:1/16:0 for SCD. 

### 2.3. Assessments of Cardiometabolic Risk Factors 

Body mass index (BMI) was calculated as the ratio of body weight (in kg) to height (in m) squared. Concentrations of cholesterol and triglycerides were measured in serum and in isolated lipoprotein fractions by enzymatic techniques utilizing Instrumentation Laboratories (IL) Test Cholesterol Trinders’s Method and IL Test enzymatic-colorimetric method for use in a monarch apparatus (Instrumentation Laboratories, Lexington, MA, USA). High-density lipoprotein (HDL) particles were separated by precipitation with magnesium chloride/phosphotungstate. IS was directly measured in the ULSAM cohort using the hyperinsulinemic euglycemic clamp, as previously described [11]. In addition, indices of insulin resistance (HOMA-IR) were assessed in both ULSAM and PIVUS as calculated using fasting concentrations of plasma glucose and insulin.

### 2.4. Preparation of Genotype Data

Genotyping was performed using the Illumina OmniExpress and Illumina Metabochip in PIVUS and Illumina Omni2.5M and Illumina Metabochip in ULSAM. General sample exclusion criteria included: (1) genotype call rate <95%; (2) heterozygosity >3 SD from mean; (3) gender discordance; (4) duplicated samples; (5) identity-by-descent match; and (6) ethnic outliers. General single nucleotide polymorphism (SNP) exclusion criteria of genotyped data before imputation included: (1) monomorphic SNPs; (2) Hardy-Weinberg equilibrium (HWE) *p*-value <1 × 10^−6^; (3) genotype call rate < 99% (SNPs with minor allele frequency (MAF) < 5%) or < 95% (SNPs with MAF ≥ 5%); (4) MAF < 1%. In ULSAM, for Omni2.5, further SNP exclusions were made if a SNP had large position disagreements, did not map in the genome, mapped more than once in the genome or had bad probe assays.

In PIVUS, 949 out of 982 samples passed QC for the OmniExpress; and Metabochip with the exclusions listed in Appendix A. The genotyped data in PIVUS used in the present study consisted of 738,879 SNPs after QC. In ULSAM, 1179 out of 1216 samples passed QC for the Omni2.5 and Metabochip, with the exclusions listed in Appendix A. The genotyped data in ULSAM consisted of 1,621,833 SNPs after QC. Imputation was performed for the quality-controlled genotype data of each cohort with IMPUTE v.2.2.2 using haplotypes from the 1000 Genomes, March 2012 release (multi-ethnic panel on NCBI build 37 (b37)). Population substructures in the genotype data were captured using multidimensional scaling (MDS) of a genetic relationship matrix (genome file) constructed on the basis of linkage disequilibrium (LD)-pruned SNPs in PLINK 1.07 [12].

### 2.5. Statistical Analysis

Estimated enzyme activity levels were normalized using Blom’s inverse normal transformation. Regression analyses of genetic variants and estimated desaturase activities were adjusted for the first two principal components of the MDS analysis. All regression analyses of the autosomes in ULSAM and PIVUS were performed separately in each study and then combined in sample size-weighted Z-score meta-analysis assuming fixed effects in the software METAL. [13] The same settings were used in the analysis of the X-chromosome as for the autosomes, with the exception that the analyses in PIVUS were stratified on gender before combining the results in the meta-analysis.

To identify single common variants associated with the estimated enzyme activity in CE, a genome-wide association analysis was performed using the score-based test in SNPTEST 2.4.1. (manufacturer, city, country) [14] Common variants (MAF ≥ 5%) available in both studies and with an information quality metric ≥ 0.4 were included in the analyses. A *p*-value < 5 × 10^−8^ was considered to be genome-wide significant in these analyses. Further, to identify independent variants in each locus associated with enzyme activity levels, a forward selection conditional analysis was performed, where independent signals were considered down to a *p*-value of ~1 × 10^−5^. In addition, genome-wide association analyses of the estimated enzyme activity in AT were performed in ULSAM only.

A lookup was performed for the significant SNPs in the single variant analyses using literature and publicly available databases including RegulomeDB version 1.1 (http://www.regulomedb.org/) [15], GTEx (http://www.gtexportal.org/home/) [16], Metabolomics GWAS server (http://mips.helmholtz-muenchen.de/proj/GWAS/gwas/) [5,6] and PhenoScanner (http://www.phenoscanner.medschl.cam.ac.uk/phenoscanner) [17]. 

Associations of identified variants with cardiometabolic risk factors including triglycerides, HDL-C, BMI and HOMA-IR were assessed by linear regression in ULSAM and PIVUS separately and subsequently meta-analyzed using sample size-weighted fixed effects models. Similarly, associations between the same variants and M-value determined by hyperinsulinemic euglycemic clamp were assessed in ULSAM using linear regression models. For these associations between desaturase-associated loci and cardiometabolic risk factors, false discovery rate was used to correct for multiple testing [18].

## 3. Results

The clinical characteristics of individuals with available genotype and cholesterol ester fatty acid data in ULSAM (*n* = 564) and PIVUS (*n* = 889) are shown in Table 1. In addition, a number of men (*n* = 783) in ULSAM also had data available on genotype and desaturase activity assessed in adipose tissue. The correlation between the estimated enzyme activity in CE and AT was low to moderate for D5D (*r* = 0.36, *p* < 0.0001), D6D (*r* = 0.10, *p* = 0.098) and SCD (*r* = 0.40, *p* < 0.0001).

### 3.1. GWAS of Desaturase Activity

Quantile-quantile plots of *p*-values from the single variant association test of the enzyme activities showed no systematic deviation from the null (data not shown). Variation in one locus (fatty acid desaturase 1, *FADS1*) was associated with D5D and variants in or near three loci were associated with D6D (fatty acid desaturase 2, *FADS2*; pyridoxal-dependent decarboxylase domain containing 1, *PDXDC1*/N-terminal asparagine amidase, *NTAN1*; and near melanocortin 4 receptor, *MC4R*) (Appendix A). The significant lead variant for CE-D5D was also significant when analyzed in AT (Figure 1), unlike D6D, where no signal could be seen in AT (Table 2). No locus was significantly associated with CE-SCD (Appendix A) but one variant close to *SCD*, in polycystic kidney disease 2-like 1 (*PKD2L1*), was significantly associated with AT-SCD, which was analyzed in ULSAM (Table 2). The same direction of effect was seen for CE-SCD, though the association was weaker (P = 1.6 × 10^−4^). 

By conditional association tests of the *FADS1-FADS2-FADS3* region, in a forward selection approach, two SNPs (rs174549, rs968567) were independently associated with CE-D5D and two SNPs (rs138194593, rs2072113) were independently associated with CE-D6D (Appendix A). The four SNPs were not in strong LD (R2 ≤ 0.40) with each other.

### 3.2. SNP Lookup

Database searches revealed that the variants independently associated with D5D activity are located in transcription factor binding regions (cited in RegulomeDB [15]) and have been associated with expression of *FADS1* and *FADS2* in diverse tissues (cited in GTEx [16]). In previous GWAS, the two SNPs independently associated with D5D (or variants in full LD) have been linked to circulating polyunsaturated fatty acids [6,19,20,21,22]. In studies utilizing candidate SNP approaches, the same SNPs have been associated with FA and ratios thereof in circulation and tissue [23,24,25,26,27].

The lead variant associated with estimated D6D activity, rs138194593, is located in an intronic region of the *FADS2* gene and has been associated with *FADS2* expression in blood from Estonian coronary artery disease patients [28]. The second independent D6D-associated SNP in the *FADS2* gene, rs2072113, or proxies in full LD have been associated to *FADS* expression (as cited in GTEx [16]) and been linked to circulating PUFA in previous GWAS [5,6,19,21]. Another variant linked with D6D activity, rs6498540, is located in *PDXDC1* and has been associated with circulating PUFA and FA ratios [6,21].

In addition, rs6498540 is in perfect LD with rs4500751, a SNP in a transcription binding region (as cited in RegulomeDB [15]) close to *PDXDC1* and *NTAN1* that has been associated with circulating PUFA and ratios thereof in previous GWAS [5,6,29]. This SNP is also located 300 kb from *PLAG10*, a gene involved in phospholipid metabolism, possibly with a fatty-acid specific mechanism [29]. The top SNP of the third locus associated with D6D activity, rs9957425, is not in strong LD with other SNPs (*r*^2^ < 0.32) and is not likely in a transcript factor binding region but may affect epigenetic modifications (as cited in RegulomeDB [15]). It has not yet been associated with circulating FA or FA ratios and has not been strongly associated (*p* ≥ 0.02) with any traits in the GWAS included in the Metabolomic GWAS scanner [5,6] or PhenoScanner [17]. The variant is located 576 kb downstream of *MC4R,* in close proximity to a region strongly associated with diabetes and related traits.

The variant associated with AT-SCD, rs603424 in the *PKLD1* gene, is located 31 kb from the SCD gene [29] and has previously been associated with adipose SCD expression and circulating saturated fatty acids, monounsaturated fatty acids and ratios thereof [5,29,30].

### 3.3. Associations of Identified Loci with Metabolic Traits

After correcting for multiple testing, the lead SNP at one of the loci significantly associated with estimated D6D activity, rs9957425, was associated with BMI (*p* = 7.4 × 10^−4^) and plasma triglycerides (*p* = 2.1 × 10^−3^) in a meta-analysis of ULSAM and PIVUS data (Table 3). The same locus was also associated to M-value in ULSAM (*p* = 3.8 × 10^−3^). Associations of the other loci with BMI, HOMA-IR, M-value plasma HDL, triglycerides, or M-value were not evident after correcting for multiple testing (Table 3). All significant loci from the single variant analysis, except rs603424 (*PKD2L1*, close to *SCD*), were associated with HDL cholesterol or triglycerides (*p* < 0.05) in a large meta-analysis of lipid values (Appendix A) [31]. In additional lookups of published data, no significant associations were seen for HOMA-IR [4] and only rs6498540 showed some evidence of association for BMI [32]. 

## 4. Discussion

In the present study, five loci were associated at a genome-wide significant level with estimated activities of D5D, D6D, or SCD in the two population-based Swedish cohort studies ULSAM and PIVUS. One of these loci, downstream of *MC4R*, was additionally associated to BMI and triglycerides in the combined study populations; and to M-value in ULSAM, after correcting for multiple testing. 

Most variants identified in the present study or proxies in perfect LD have been associated with circulating FAs, ratios thereof or *FADS* gene expression in previous GWAS [5,6,19,20,21,22,28,29,30]. However, one novel variant near the *MC4R* gene was identified. Polymorphisms downstream of the *MC4R* gene are among the strongest genetic determinants of BMI and they have been associated with food preference, IS and HDL [33]. Estimated D6D activity has likewise previously been positively associated to obesity [34,35] and it can be speculated that the observed associations of rs9957425 with desaturase activity, BMI and IS are due to MC4R-mediated effects on food and fatty acid intake, which could influence FA proportions and thereby the FA ratio used for estimating D6D activity.

As expected, the strongest associations with estimated D5D and D6D activities in ULSAM and PIVUS were observed with variants mapped to the *FADS* cluster, which supports the use of FA ratios as estimates of desaturase activity. Polymorphism in FADS encoding genes may be directly linked to IS and T2D [36]; however, identification of such associations could be hampered by high LD in the *FADS* region and pleiotropy of the *FADS* genes. Most genetic variants associated with FADS activity were linked to blood lipid levels in meta-analysis of published GWAS [31], supporting the relationships between FADS activity and development of metabolic syndrome [37]. Thus, it is a challenge to disentangle the potential direct role of *FADS* gene variants on IS, from that of closely related metabolic disorders such as triglycerides and HDL.

The correlations between the estimated enzyme activities in CE and AT observed in the present study confirm results from a previous study in which activities of D5D and SCD in AT and serum were correlated in Swedish men and women [35]. In that study, D6D activity estimated in AT was correlated with that estimated in phospholipids but not non-esterified serum FA. Another study reported that estimated enzyme activities of D5D, D6D and SCD in serum (calculated from total FA in serum) were highly correlated with corresponding activities in liver tissue but not in AT [38]. Similarly, enzyme activities of fatty acid desaturases estimated in different plasma lipid fractions are not ubiquitously correlated, though enzyme activities estimated in cholesteryl esters used in the present study are considered to reflect hepatic fatty acid desaturation [35,39]. 

Associations between gene expression and enzyme activity of SCD have been observed in liver [40], AT [41] and brain [42]. It should be noted that SCD gene expression may differ in different adipose tissue depots and we have reported that SCD gene expression was correlated with estimated SCD activity in subcutaneous but not in visceral adipose tissue [43].

Similar associations for D5D have been observed in brain [42] and for D6D in liver [40]. Although product-to-substrate ratios are indirect measurements of enzyme activity and their accuracy has been questioned [44,45], FA ratios correlate with direct enzyme activity measurements (by isotope tracer) [46]. The FA composition of CE is regulated in the liver and plasma, while the composition in AT is influenced by adipose metabolism [47]. For example, FA are released from AT by lipase-catalyzed lipolysis and diverse mobilization of individual FA from AT that could affect the ability of FA ratios to estimate enzyme activity in AT [48]. Hence, as FA ratios in CE and AT represent desaturase activities in various tissues (i.e., liver, plasma and adipocytes) they may be affected by diverse genetic determinants.

A major strength of the present study is the availability of data on FADS activity estimated in both AT and serum and to our knowledge this is the first GWAS of adipose FADS activity. Furthermore, IS was assessed both by the gold standard methodology hyperinsulinemic euglycemic clamp and calculated using HOMA-IR. The combination of the two cohorts, ULSAM and PIVUS resulted in dataset consisting of men and women with measured FA composition.

The relatively small sample size of the two cohorts utilized is a limitation of the present study. Also, data on adipose FADS activity and M-value were only available in one of the cohorts, ULSAM. As the two cohorts consist of individuals from the same geographical location, there is a possibility that the two study population are too homogeneous and thus the possibility to identify associations between genes and fatty acid metabolism are hampered by low genetic variation. Further, the generalizability to other ethnicities is unknown. 

Our findings have implications for future research. First, our findings support the use of FA ratios as indirect estimates of desaturase activity, given that the variants most strongly associated with FA ratios were located in or near desaturase encoding genes. However, certain lipid fractions may be less suitable to assess desaturase activity by FA ratios as suggested by the inconsistency in loci associated with D6D and SCD in different fractions. In addition, our findings warrant further evaluation of rs9957425, associated with both desaturase activity and cardiometabolic traits.

## 5. Conclusions

In conclusion, the activities of FADS estimated in CE and AT were associated with variants in or near five independent loci (*FADS1*, *FADS2*, *MC4R*, *PDKL1* and *PDXDC1*). One of the loci (*FADS1*) was associated with FADS activity in both CE and AT. One variant associated with estimated D6D activity (rs9957425 near *MC4R*) was additionally was associated to BMI, TG and intravenously assessed IS. Activities of D5D and SCD estimated in CE and AT were correlated, while no correlation was observed for D6D activity estimated in the different tissues.

## Figures and Tables

**Figure 1 nutrients-10-01791-f001:**
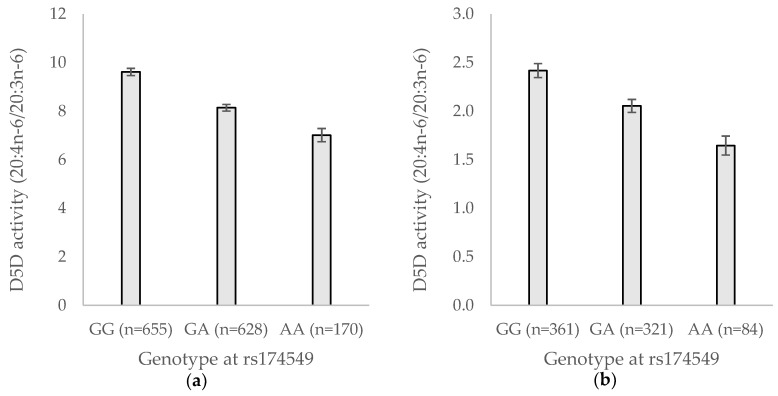
Estimated δ-5-desaturase (D5D) activity in cholesterol esters (**a**) and adipose tissue (**b**) by genotype at rs174549. Values are means and error bars represent 95% confidence intervals of means. D5D activity was estimated as the ratio of arachidonic acid (20:4n−6) and dihomo-gamma-linolenic acid (20:3n−6).

**Table 1 nutrients-10-01791-t001:** Clinical characteristics of individuals in ULSAM and PIVUS with genotype data and estimated desaturase activity in cholesterol esters. ^1^

	ULSAM (*n* = 564)	PIVUS (*n* = 889)
Age	71.3 (0.4)	70.2 (0.2)
Women (%)	0	49
BMI (kg/m^2^)	26.3 (3.4)	27.0 (4.4)
Antihypertensive treatment (%)	37	31
Total cholesterol (mmol/L)	5.8 (1.0)	5.4 (1.0)
HDL cholesterol (mmol/L)	1.3 (0.3)	1.5 (0.4)
Triglycerides (mmol/L)	1.5 (0.8)	1.3 (0.6)
Lipid lowering treatment (%)	9.4	15.9
Fasting plasma glucose (mmol/L)	5.8 (1.5)	6.0 (1.8)
Glucose disposal, M (mg/kg/min)	5.2 (2.1)	N/A
Diabetes treatment (%)	6.5	6.5
Never smokers (%)	40	48
Previous smokers (%)	40	41
Current smokers (%)	20	11

Values are mean (SD) or percentage. ^1^ PIVUS, Prospective Investigation of Uppsala Seniors; ULSAM, Uppsala Longitudinal Study of Adult Men; HDL, high-density lipoprotein; BMI, body mass index; M, in vivo insulin-mediated glucose disposal.

**Table 2 nutrients-10-01791-t002:** Common variants (MAF ≥ 5%) with *p* < 1 × 10^−8^. ^a^

Desaturase	Gene	Lead SNP	Chr:Position (b37)	EAF	Effect Allele/ Other Allele	Serum			Adipose Tissue		
						Direction	*p*	*n*	Direction	*p*	*n*
D5D	*FADS1*	rs174549	11:61571382	0.33	A/G	−	1.9 × 10^−70^	1448	−	1.1 × 10^−27^	766
D6D	*FADS2*	rs138194593	11:61620703	0.63	CTCTT/C	+	3.0 × 10^−67^	1448	−	7.9 × 10^−1^	611
D6D	*PDXDC1*	rs6498540	16:15130594	0.71	A/G	+	4.8 × 10^−8^	1448	−	2.9 × 10^−1^	611
D6D	near *MC4R*	rs9957425	18:57462103	0.61	T/C	+	3.7 × 10^−8^	1448	−	9.0 × 10^−1^	611
SCD	*PKD2L1*	rs603424	10:102075479	0.14	A/G	−	1.6 × 10^−4^	1453	−	2.2 × 10^−19^	783

^a^ A, adenine; C, cytosine; Chr, chromosome; b37, NCBI build 37; D5D, δ-5-desaturase; D6D, δ-6-desaturase; EAF, effect allele frequency; *FADS1*, fatty acid desaturase 1; *FADS2*, fatty acid desaturase 2; G, guanine; *MC4R*, melanocortin 4 receptor; *PDXDC1*, pyridoxal-dependent decarboxylase domain containing 1; *PKD2L1*, polycystic kidney disease 2-like 1; SCD, Stearoyl-CoA desaturase; T, thymine.

**Table 3 nutrients-10-01791-t003:** Associations of significant desaturase loci with BMI and metabolic traits. ^a^

Gene	rs ID ^b^	Chr:Position (b37)	Effect Allele	EAF	ULSAM + PIVUS Meta-Analysis (*n* = 1453)								ULSAM (*n* = 564)	
					HDL Cholesterol		Triglycerides		BMI		HOMA-IR		M-Value	
					Direction	*p*	Direction	*p*	Direction	*p*	Direction	*p*	Direction	*p*
*FADS1*	rs174549	11:61571382	A	0.33	−	2.0 × 10^−2^	+	1.8 × 10^−1^	+	5.4 × 10^−1^	+	6.9 × 10^−1^	−	7.6 × 10^−1^
*FADS2*	rs968567	11:61595564	T	0.15	−	2.4 × 10^−1^	+	5.5 × 10^−1^	+	1.1 × 10^−1^	+	5.6 × 10^−2^	−	2.1 × 10^−1^
*FADS2*	rs138194593	11:61620703	CTCTT	0.63	+	9.3 × 10^−3^	−	4.2 × 10^−1^	−	1.8 × 10^−1^	−	7.9 × 10^−1^	+	5.3 × 10^−1^
*FADS2*	rs2072113	11:61604967	T	0.17	−	7.6 × 10^−2^	+	3.9 × 10^−1^	−	7.3 × 10^−1^	−	2.5 × 10^−1^	+	5.8 × 10^−1^
*PDXDC1*	rs6498540	16:15130594	A	0.71	+	7.8 × 10^−1^	−	8.5 × 10^−1^	−	4.2 × 10^−1^	+	3.7 × 10^−1^	+	7.6 × 10^−1^
near *MC4R*	rs9957425	18:57462103	T	0.61	−	2.2 × 10^−1^	+ ^c^	2.1 × 10^−3^	+ ^c^	7.4 × 10^−4^	+	7.1 × 10^−2^	− ^c^	3.8 × 10^−3^
*PKD2L1*	rs603424	10:102075479	A	0.14	+	8.4 × 10^−1^	+	5.2 × 10^−1^	+	8.8 × 10^−1^	+	8.8 × 10^−1^	−	7.6 × 10^−2^

^a^ Chr, chromosome; b37, NCBI build 37; EAF, effect allele frequency; *FADS1*, fatty acid desaturase 1; *FADS2*, fatty acid desaturase 2; *PDXDC1*, pyridoxal-dependent decarboxylase domain containing 1; *MC4R*, melanocortin 4 receptor; *PKD2L1*, polycystic kidney disease 2-like 1; PIVUS, Prospective Investigation of Uppsala Seniors; ULSAM, Uppsala Longitudinal Study of Adult Men; HDL, high-density lipoprotein; BMI, body mass index; HOMA-IR, homeostasis model assessment of insulin resistance. ^b^ Reference SNP ID number. ^c^ Significant association after adjustment for multiple testing.

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
