# Peer review of "Genome-Wide Association Studies of Estimated Fatty Acid Desaturase Activity in Serum and Adipose Tissue in Elderly Individuals: Associations with Insulin Sensitivity"

_nutrients, 2018, doi:10.3390/nu10111791_

Round 1
Reviewer 1 Report
The manuscript entitled: “Genome-Wide Association Studies of Estimated Fatty Acid Desaturase Activity in Serum and Adipose Tissue in Elderly Individuals: Associations with Insulin Sensitivity” by Matti Marklund et al. describes SNPs in relation to desaturase activities. The study is well done and has an interesting/novel hypothesis. However, a few specifications are needed to better understanding of the reader.
Both circulating fatty acid composition and estimated FADS activities have previously been associated with insulin sensitivity (IS) and incidence of type 2 diabetes (T2D), although the mechanisms underlying these relationships have not been fully determined. Please explain how these activities/SNPs affect IS in more details.
You mention in methods that you have measured HOMA-Beta; however, no results are shown. Please add. Similar, in introduction you state that IS will be used. However, other metabolic traits were also analysed. Please adjust.
Results for other genes are shown in Table 2. However, we do not know the pertinence of these genes with Desaturases ratio. Please elaborate in introduction (and abstract e.g. PKDL1, MC4R).
Hence, as FA ratios in CE and AT represent desaturase activities in various tissues they may be affected by diverse genetic determinants. Can you elaborate how this may affect the metabolism?
Conclusions: Please state the potential utility of the results (in research or in clinic).
Author Response
The manuscript entitled: “Genome-Wide Association Studies of Estimated Fatty Acid Desaturase Activity in Serum and Adipose Tissue in Elderly Individuals: Associations with Insulin Sensitivity” by Matti Marklund et al. describes SNPs in relation to desaturase activities. The study is well done and has an interesting/novel hypothesis. However, a few specifications are needed to better understanding of the reader.
1. Both circulating fatty acid composition and estimated FADS activities have previously been associated with insulin sensitivity (IS) and incidence of type 2 diabetes (T2D), although the mechanisms underlying these relationships have not been fully determined. Please explain how these activities/SNPs affect IS in more details.
Thank you for the comment. As we stated before, the mechanisms are not fully understood. It has however been suggested that the effects of desaturase activities are mediated by alterations in FA compositions. We have extended the paragraph to describe some of the suggested mechanisms (lines 51-55).
2. You mention in methods that you have measured HOMA-Beta; however, no results are shown. Please add. Similar, in introduction you state that IS will be used. However, other metabolic traits were also analysed. Please adjust.
Thank you for noticing this. We have now removed the text regarding HOMA-Beta. In addition, we have also reviewed the text throughout the manuscript to ensure that insulin sensitivity is abbreviated IS. Hopefully, these alterations will help avoid any confusion.
3. Results for other genes are shown in Table 2. However, we do not know the pertinence of these genes with Desaturases ratio. Please elaborate in introduction (and abstract e.g. PKDL1, MC4R).
Given our approach to conduct a GWAS instead of evaluating candidate SNPs and the indirect measurement of desaturase activities (using FA ratios), there are countless potential loci associated with the FA ratios. We have made an alteration to the introduction about genetic variants associated with FA ratios not necessarily being located in genetic regions not related to desaturases (lines 59-60). In the results and discussion paragraphs, we describe in detail the identified loci in relation to FA desaturation.
4. Hence, as FA ratios in CE and AT represent desaturase activities in various tissues they may be affected by diverse genetic determinants. Can you elaborate how this may affect the metabolism?
We have now extended the discussion to clarify that FA composition in CE is regulated by enzymes in liver and plasma, whereas metabolic processes in AT influence the FA composition there, and thus FA rations estimated in the different fractions are likely to be influenced by difference genetic factors (lines 267-272).
5. Conclusions: Please state the potential utility of the results (in research or in clinic).
Thank you for this suggestion. We have extend the discussion with a paragraph regrinding the impact of our findings (lines 284-289).
Reviewer 2 Report
In the abstract, authors wrote, "Variants at the FADS1 were strongly associated with D5D in both CE (P=1.9×10-70) and AT (P=1.1×10-27)". I recommend that this strong association should also be shown visually to use any figure. Because I think D5D activity is continuous variable, I wonder if scatter plot might be good.
Author Response
In the abstract, authors wrote, "Variants at the FADS1 were strongly associated with D5D in both CE (P=1.9×10-70) and AT (P=1.1×10-27)". I recommend that this strong association should also be shown visually to use any figure. Because I think D5D activity is continuous variable, I wonder if scatter plot might be good.
Thank you for this suggestion. We have now included a figure (Figure 1) presenting D5D activity estimated in cholesterol ester and adipose tissue by genotype at rs174549 (the top SNP of D5D activity).
1. Novelty: I haven’t seen any papers to investigate the association between desaturase activities and GWAS data. Therefore, I believe the finding is precious and bran- new.
Thank you for acknowledging the novelty of our research.
2. Experimental method I think the experimental methods are adequate.
Thank you.
3. Expression or significance The authors should consider degree (or strength) of associations. I know only P value is evaluated in GWAS study, but values of desaturase activities are im portant. Desaturase activities are correlated with insulin resistance (previous study of authors) and visceral fat accumulation. Main findings are significant associations between serum D5D & FADS1(P=1.9x10 -70), and serum D6D & FADS2(P=3.0x10-67). Moderate to strong associations are important, clinically. For example, readers will estimate the different if authors show variations of D5D activity values in the effect and other alleles, respectively. Authors can make most adequate figures to show how important their findings. When a new figure make me to feel moderate to strong association (or defferen ce), I believe this article has a strong impact on readers.
We agree with the reviewer of the importance of magnitude of associations and in order to comply with the reviewers comments we have added Figure 1, which (as described above) presents genotype-specific D5D activity estimated in cholesterol esters and adipose tissue.